# Considerable Factors for Domain Model Learning in Planning

**Rabia Jilani** and **Rubiya Reba**
University of Huddersfield, Queensgate, UK

### Abstract

Learning for planning has extensive aspirations to aim at. Our primary focus area of review in this paper is the knowledge engineering of domain models to feed planners. This concerns with learning and representation of knowledge about operator schema, discrete or continuous resources, processes and events involved. With the increased problem complexity on the continuum of planning from classical towards full scope, synthesizing expressive and intensive planning domain models by hand become more challenging, time-intense and error-prone. Our intended contribution is to provide a broader perspective on the range of research in the domain model learning area and specification of interdependent AP dimensions which can affect the developmental decisions of learning methods/techniques.

We map the amplitude of recently created domain model learning methods on five considerable dimensions. The dimensions to underpin the development of a learning system include the type of planning paradigm, representation languages, learning at various planning stages, learning systems and sources and the extent of learning that takes place. We also identify considerable issues for future work.

## Introduction

Automated Planning (AP) is one of the most prominent AI challenges. It is the process of finding a procedural course of action through explicit deliberation process to reach a pre-stated objective in the form of goals while optimizing overall performance. Planning is a pivotal task that has to be performed by autonomous agents.

In order to perform automated reasoning, planning techniques require formal specification of application knowledge to be encoded in the form of Domain Models (DM). In the action-centred view of problem representation, DM encodes the domain knowledge in the form of actions that can be executed together with relevant action properties and features. The correctness of the planner reasoning depends fundamentally on the quality of the domain knowledge that otherwise can prove catastrophic. In the complex domain scenarios, planners also use manually encoded or automat-

ically learned domain-specific control knowledge (in addition to DM) to guide planner search and cater to scalability issues. Most planners define control knowledge separately from DM to support different representations. Here we only focus on DM learning aspect.

Acquisition of DM from scratch is an exceedingly difficult and expensive task that is usually done manually. KE for planning DMs using Machine Learning (ML) techniques is considered as a paramount for empowering autonomous learning systems with the capacity to fill implicit human knowledge gaps and errors, requiring least human intervention. The area of ML application to DM learning systems has received active research attention in recent years but did not make as much stride as the learning of control knowledge. Not enough depth of research into the area has made it an ad-hoc process, where the skills of knowledge engineers significantly influence the quality of the resulting planning application and the accuracy of DM still counts as a bottleneck for AP (McCluskey, Vaquero, and Vallati 2017).

In this study, our intention is to provide a broader perspective on the range of research in the area of Learning DMs for AP. We include various factors that make AP a multi-dimensional area for learning researchers and map the amplitude of recently created learning methods to these dimensions. Currently, most of these DM learning systems are offline learners that target a very narrow range of diverse AP domain features and exploit a specific source of input i.e. plan traces with or without intermediate states using Inductive generalization technique to learn classical DM (Jilani et al. 2014). The target readers of the paper are the students and interested researchers. The basic motive is to draw their attention at the current research focus, the overlooked issues and the gaps which given more attention has a potential for exploration and research. We assume that the reader would have insights into the planning and learning baseline literature.

In the next section, we look at learning DMs from different considerable factors including planning paradigms to learn domain models for, representation mechanism, learning/improving DMs at different planning stages, the types of learning sources/inputs and the extent of learning as an outcome. We then discuss the types of learning techniques and algorithms exploited by the learning systems. Towards the end, we infer some effective and worthwhile venues for

further efforts in the area.

## Considerable Factors for development of DM Learning system

Following planning dimensions are related to learning and dependent on each other in the number of ways and compatibility scenarios. Certainly, this is by no means an exhaustive list of considerable factors and techniques that may affect the learning-planning duo. Among other prospective dimensions that could have been considered include; structural type and correctness of input and learning drivers or motivators (data-driven or knowledge-driven) for learning.

The operationality of the automatically learnt domain models depends upon what actions do and how they do it. On the spectrum of working domain models, Descriptive model of actions identify 'what' part of the deliberation by its awareness of the resulting set of states after executing a particular action. In comparison, Operational models describe 'how' to carry out task through actions by demonstrating awareness of how to perform an action and achieve the target effects (Ghallab, Nau, and Traverso 2016). On the scale of the deliberation process, descriptive models are higher level abstract models which focus on action effects. This is useful for reasoning under situations where planning information if not available on the run time as hierarchy and abstraction make planning more efficient. Moving on the scale towards operational model to address the real-world scenarios and to include ways to respond to exogenous events, is computationally very complex yet employ fundamentally identical learning techniques.

To explain the difference between Descriptive and Operational DMs, (Ghallab, Nau, and Traverso 2016) quotes an example of modelling for an agent to take a book from the library book shelve. On planning time, a descriptive model will abstract away the details like available space on the sides or top of the book to insert fingers and extract book from the shelf. It will instead focus on how to reach the book and whether hand/gripper needs to be free to execute the action. On the other and, for Operational model the scenario becomes way more complex with determining precisely the position of book and hand in addition to feasibility in action and motion sequencing to lift the books.

From the collection of work to date, learning and planning approaches are compared to similar or seminal research while the research to reveal interrelations and interdependencies are least addressed. We discuss five interrelated dimensions to stretch the consideration pane when working towards DM learning for model-based planning and may help in identifying open venues for future work. Following is the brief discussion on each of these dimensions in the perspective.

### Planning Paradigms and Problem Types

Since the classical planning paradigm was proposed in the 50s, considerable progress has been made in the development of planning techniques and domain-independent planners. Currently, the dimensions of planning and so the underlying domain model design is wide-ranging. The continuum of the categories from "classical" to "full scope" planning depends on the nature of the environment in which planner has to do deliberation (Zimmerman and Kambhampati 2003). Deliberation assumptions broadly depend upon seven features, including state observability, action determinism, world dynamics, action duration, the scope of the state, attainment of goal and time and concurrency factor. The classical planning has the most rigid assumptions while the assumptions relax progressively towards full scope planning on the continuum. In this section we clarify how the DM design and dynamics change over the continuum of the planning types.

**Classical Planning:** Classical planning reason about static and fully known environment that has a finite and discrete set of states and fully deterministic actions effects. The actions in classical planning occur instantaneously and do not consider any exogenous changes that are expected in the environment. All these factors modelled in the form of a discrete DM for a classical planning problem often differ with the real world in which plans must be executed. This makes it more of an abstract and descriptive model which is far from a continuous operational model with the ability to response exogenous events.

The states in a DM are a valuation of the variables from the domain. Two variations of languages for expressing classical DMs include the one with boolean state variables and the other with multi-valued state variables that take values from a finite domain. STRIPS language to encode classical DMs is based on the boolean variables that put into effect classical state model. Several languages that include richer syntax and finite-domain non-boolean state variables instead of the straight-forward propositional encoding include (McDermott et al. 1998)(Bäckström and Nebel 1995)(Gerevini et al. 2009)(Helmert 2009).

**Metric Planning:** Logically based on classical planning except that it reasons in the presence of numeric variables without a bound on the number of values i.e. infinite variables. Numeric variables in metric DM usually represent consumable resources e.g. fuel, time, energy, capacity etc. Semantically being close to classical planning, a metric domain can be mapped by applying a finite bound on the range of multi-valued numeric variables.

**Temporal Planning:** Unlike classical planning DM where time is considered implicit with instantaneous action, temporal planning DM explicitly include duration in order to model concurrency in both independent and overlapping interactive actions. In addition to usual pre and post conditions in action definition, temporal DM also mentions numeric constraints (relative or absolute) and events to locate action execution in time. Generally, the objective of modelling temporal constraints is a minimization of the plan makespan. Time can explicitly be represented in the DM using state-oriented or time-oriented representation(Ghallab, Nau, and Traverso 2016). PDDL 2.1 (Fox and Long 2003) models temporal DM with the group of propositions and a vector of numeric variables to be manipulated by the actions based on their preconditions. Temporal planning closely relates to Scheduling.

**Hybrid Planning:** A hybrid system is one in which there are both continuous control parameters and discrete logical modes of operation. Advances in classical and temporal planning have led to a possibility to plan in continuous change. Hybrid planners and DMs tend to contribute towards indispensable value in building autonomous reasoning systems. To reason about non-linear continuous dynamics in addition to conventional problem solving, the underlying DM of the world needs to be more expressive.

DM for hybrid planning include object types, action definition (that can be conditional) and the continuous (numeric) effects affecting the typed variables. Hybrid DM also contain exogenous events and processes that certainly executes regardless of the action taken. Processes model continuous changes in the environment with time and execute when its preconditions become true. Solar rover domain can be a good example; with the Rovers, chargeable Batteries and the rover Engines etc be the considerable types. Drive or Recharge batteries would be the actions, while sunrise and sandstorm would be the events and process. Execution instance of continuous numeric effects may occur as a result of some action effects e.g. when rover has to change its direction, move, align batteries and transmit data, all of these actions would have separate continuous effects on the battery charging process of the rover.

AP community faces significant challenges in the construction of hybrid DMs. One main exhaustive challenge includes expressivity of the model elements for continuous non-linear change in the target problems. In addition, it is difficult to generalise from continuous change. To date, the only viable approach towards hybrid planning and learning is PDDL+ planning via discretisation. Several authors are working towards the hybrid planning domain using piecewise linear function and dealing with continuous change by discretising time and solving it as a discrete time problem (Denenberg and Coles 2018b) with scalability as a major issue to face. None of the hybrid planning solvers can perform with real-world problems except after choosing the solver to fit the purpose, or by deliberately modelling in such a way so as to make the problem solvable in a realistic time. Based on the fact that thorough and robust DM is rare, the development of hybrid DM learning systems is in its infancy.

Prominent examples of continuous and dynamic domains in large scale operations include urban traffic management and control (UTMC) domain (McCluskey and Vallati 2017), Reservoir Control (Yeh 1985), Heating, Ventilation and Air Conditioning (HVAC) (Erickson et al. 2009) and Navigation (Faulwasser and Findeisen 2009). Such scenarios motivate the need to automate or semi-automate the hybrid DM learning process.

## Representation Language

Expressively and practicability are generally the two main considerations for knowledge representation mechanism. In AP the key purpose of explicit knowledge representation language for the formulation DM is to be able to reason with it and infer new knowledge from it in the form of action plans. There is no one knowledge representation approach just like the reasoning approach that has combined properties for all types and level of deliberation problems. Similarly, there is no single highly specialized knowledge representation to cover a specialized area of learning DMs. A well-chosen representation language should explicitly model every action effect the system might confront (Arora et al. 2018a). (Riddle, Holte, and Barley 2011) proves that the representation mechanism exploited makes an extensive difference to the planner's ability to solve a problem by exploring six different representations of the Blocks World Domain.

To cover diverse terrain of domain modelling approaches and the kind of symbolism needed corresponding to respective planning types, this section discusses the most commonly used and accepted knowledge representations that work well with their corresponding planning paradigm.

The term STRIPS is substantially used to refer to formulations for classical planning. STRIPS was developed at Stanford in early 70s, for the first intelligent robot. STRIPS encoding of DM includes state-transition function whereas the states are represented as first-order predicates. First order logic (or predicate logic) is considered as the most expressive representation formalism for reasoning systems in a dynamic environment. Based on its restrictive features, STRIPS was proposed to avoid the frame problem complexity as an alternate representation mechanism for situation calculus. Being a limited language in terms of semantic features, reasoning and expressive power, STRIPS was augmented by PDDL (McDermott et al. 1998) in order to acquire common formalism and standardization for planning problems.

PDDL family of planning domain description languages is based on STRIPS (and partly ADL) assumptions while supporting more expressive features and representations. PDDL express DM based on types, predicates, constants and operator definitions. Based on the feature space provided by PDDL, DM representation has progressed from representing toy problem to being the foundation of autonomous behaviour with the evolution of AP.

The PDDL domain modelling features and elements have been extended beyond some of the traditional restrictions of classical planning (for the environment to be deterministic, fully observable and static etc) (PDDL 1.2). It includes numeric variables for numerically expressive resources and durative actions to express the temporal structure of actions (PDDL 2.1 (Fox and Long 2003)). As an extension of PDDL 2.1 for probabilistic domains models with stochastic state transitions, PPDDL (Probabilistic PDDL) (Younes et al. 2005) was proposed. PDDL 3.0 include hard and soft constraints (Gerevini and Long 2006) and object-fluents (PDDL3.1 (Helmert 2008)).

RDDL (Sanner 2010) turned out to be very rich language by its use in IPPC-2011 (Vallati et al. 2015) (to replace PPDDL) in producing models with the factor of stochasticity, concurrency, and complex reward which is not possible with PPDDL.

PDDL+ (Fox and Long 2006) was developed to model more flexible, robust and autonomous hybrid (mixed discrete and continuous) DMs to reason about non-linear continuous dynamics with high dimensional state and action

spaces. The ability to manage interaction between the system and environmental exogenous events is the key to this PDDL extension. An advantage of using the continuous PDDL+ over the classical PDDL is its accuracy and granularity of its continuous representation (McCluskey and Vallati 2017). Several researchers have argued that the model adopted in PDDL+ is much more natural than the previously build models (McDermott 2003a)(Boddy 2003). The modelling of continuous processes has also been considered by (McDermott 2003b) and several other earlier in the knowledge representation and reasoning communities.

In PDDL+ encoding of DM, events are analogous to instantaneous actions when the event preconditions satisfy. Similarly, processes are equivalent to continuous durative actions of PDDL 2.1. Unlike action execution that happens if chosen to, processes and events automatically occur as soon as their precondition satisfies. Processes are triggered by actions or by exogenous events. The process executes based on its satisfied preconditions which lead to the execution of continuous numeric changes as process effects. So only processes contain continuous update expression while actions and events, being instantaneous, still represent discrete change. Because of the higher complexity of mixed discrete/continuous planning, very few PDDL+ planners are available and this currently counts as a disadvantage of using PDDL+ representation for domain modelling.

OCL (McCluskey, Richardson, and Simpson 2002), ADL (Pednault 1989), NDDL (Frank and Jónsson 2003), MA-PDDL (Kovács 2012) and OPT (McDermott 2005) are a few more planning languages not very popular for DM learning.

Mccluskey et al in (McCluskey and Vallati 2017) introduced a number of DM properties that also assist in the effective selection of representation mechanism for domain encoding. These include consistency, accuracy, completeness, adequacy and operationality with respect to the formation of planning DM and their associated problem instances in particular representation language.

## Planning Stages and potential Learning

In order to deal with real-world planning-inherent complexity, learning-augmented planning systems should be able to apprehend the environment, generate corresponding effects and enhance their performance according to the previous experience. We discuss learning-augmented planning systems from DM learning and refinement perspective. These systems can be categorized as conducting learning before the planning process starts (offline), during the plan search and execution stages (Online).

Both online and offline DM learning has pros and cons. Online learning can continuously/incrementally refine the DM in case an anomaly is detected by improving or adapting to the changes while for offline planning, the planner has to bear with the predefined version till the planning process finishes. Similarly, For online learning of DM, the overhead cost incurred for the joint planning-learning process is higher in terms of processing time and efficiency compared to offline learning (Zimmerman and Kambhampati 2003). This may also explain why online incremental DM learning has not been very popular in recent years.

The form of online learning approaches that have received active attention is for learning domain control knowledge. Unfortunately, very few learning integrated planning systems perform online learning for completion or improvement of DM. Few systems that exhibit online learning include online learning of macro-actions during planning process (Coles and Smith 2007)(Botea et al. 2005) or learning patterns for predicting `success`, `failure` or `dead-end` outcomes on plan execution to compile them into a new action model (valid for off-the-shelf planners) (Jiménez, Fernández, and Borrajo 2008) . Reinforcement learning conducts online learning through the trial-and-error visitation of states in its environment to seek optimal policy for problem goals. Reinforcement learning can be used to enhance plan quality as well as to learn the DM (begins without a model of transition probability between states) (Zimmerman and Kambhampati 2003)(Croonenborghs et al. 2007).

## Learning Systems and their Sources

This section provide insights into various learning systems that also goes beyond the boundary of learning for AP but essentially use the known ML techniques to conduct learning.

**Apprentice System:** (Mitchell, Mabadevan, and Steinberg 1990) coined the term apprentice system as an interactive knowledge-based system which comprehends knowledge by observing the users interaction with the system and analyzing the problem-solving steps. These systems capture and infer from the training examples of the user's activity and the context on which decisions involved in activity were taken. It then generalizes rules from the training examples that are comparable to the hand-generated rules. The idea has been implemented in a number of application areas. LEAP (Mitchell, Mabadevan, and Steinberg 1990) is a learning apprentice and advice system for digital circuit design that learns new tactics from its experience with user approval and rejection of its advice about circuit decomposition. ARMS system (developed in 1988) (Segre 2012) for robot Assembly planning learns from user interface where user instructs the simulated robots to perform simple tasks. (Nakauchi, Okada, and Anzai 1991)(Jourdan et al. 1993)(Tecuci and Dybala 1998)(Abbeel et al. 2008) are some more examples of apprentice systems that use various methods of interaction with the user including passive observations or querying the user to record the reason behind the particular decision made.

**Learning from Demonstration:** The source of input for such systems are generally the training logs based on the sub-cognitive skills and actions of a trainer. The AP literature also describes this method as learning by observation, demonstration, imitation or watching and also by Behavioral Cloning. Learning from fully or partially-annotated demonstrations by domain expert have been used by several systems for knowledge acquisition in robotics and task modelling (Garland and Lesh 2003)(Argall et al. 2009) but has rarely been used to learn declarative DM (Nejati, Langley, and Konik 2006) for AI planners. This is partly because for moderately complex domains, it is unfeasible for the area

expert to specify conjecture for every action, explanations for every inconsistency, and all possible effects of DM, as the performance of the learner is affected by the ability of the trainer. In order to cover for trainer's implicit knowledge gap, many systems use reinforcement learning techniques to co-operate with this type of learning. Two of the common demonstration approaches include Tele-operation and Shadowing.

**Crowdsourcing:** Recently crowdsourcing (Howe 2008) has been exploited as a novel approach for acquiring planning DMs (Zhuo 2015). Collecting a large amount of training data is not always feasible in terms of reach and cost e.g. in a situation like a military operation. Instead of collecting training examples, crownsourcing methods engage different annotators that could include various sources like domain experts, stakeholders, previous data, or experience of the general public about the domain to learn. The outcome from various annotators built as the soft constraints can later be solved using Max-sat solver to generate a domain model.

While crowdsourcing is comparatively new in learning for planning, it has been used in several planning application e.g (Zhang et al. 2012) enables a crowd to effectively and collaboratively resolve global constraints to carry out itinerary planning. (Gao et al. 2015) proposes a technique to handle the discrepancy in crowd inputs by first building a set of Human Intelligence Tasks (HITs) for values collection and then estimate the actual values of variables and feed the values to a planner to solve the problem. (Raykar et al. 2010) label training data for machine learning by crowdsourcing information from experts and non-experts. The system not only evaluates the different experts and but also gives an estimate of the actually hidden labels.

**Transfer learning:** Through Transfer Learning (TL), system exploits data from one or more source domains to improve learning performance in a different target domain in situations with like limited training data availability. Knowledge engineering through transfer learning especially for planning DMs has recently received much attention and the resulting learning technique provides a good corpus of work for interested researchers.

Pan et al in a survey on TL explain the benefit of using TL to cover the same feature space assumption of ML. Many machine learning methods work well only under a common assumption: the training and test data are drawn from the same feature space and the same distribution. When the distribution changes, most statistical models need to be rebuilt from scratch using newly collected training data. In many real-world applications, it is expensive or impossible to recollect the needed training data and rebuild the models. It would be nice to reduce the need and effort to recollect the training data especially when data is scarcely available or when it easily becomes outdated. In such cases, knowledge transfer or transfer learning between task domains would be desirable (Pan and Yang 2010). The survey also addresses the primary issues of what, when and how to transfer.

(Zhuo, Yang, and Li 2009) learns DM from plan traces by transferring useful information from other domains whose DMs are already known. The system creates a metric to measure the shared information and transfer this information according to this metric. The larger the metric is, the bigger the information is transferred. Inspired by the educational psychology of meta-cognitive reflection for better inductive transfer learning practices (WEI et al. 2018) propose a novel L2T framework for transfer learning which automatically optimizes what and how to transfer between a source and a target domain by leveraging previous transfer learning experiences. (Zhuo et al. 2008) presents t-LAMP, (transfer Learning Action Models from Plan traces) which can learn DMs in PDDL language with quantifiers. The system exploits plan traces using MLN to enable knowledge transfer. (Zhuo and Yang 2014) proposed TRAMP (Transfer learning Action Models for Planning) system, to learn DMs with limited training data in a target domain, by transferring as much of the available information from source domains by using web search on top of transfer technique to bridge the transfer gap.

**LSTM:** Long short-term memory (Hochreiter and Schmidhuber 1997) is artificial recurrent neural network (RNN) architecture used in deep learning techniques and has been successfully used in learning long range dependencies. Along with its common applications in a number of areas like connected handwriting recognition (Graves et al. 2009) or speech recognition (Sak, Senior, and Beaufays 2014), it is recently been used in learning for planning perspective.

LSTM architecture and plan generation, both effectively exhibit the same phenomena of exploiting long-range dependencies to function. The PDeepLearn (Arora et al. 2018b) is a pioneering system that maps the LSTM abilities to learn a high-quality multimodal HRI PDDL model identical to the hand-woven DM for AP. The source of learning exploited for this purpose is state-action interleaved plan traces. The PDeepLearn system narrows down the ideal model out of the candidate DMs generated, by exploiting pattern mining techniques and recurrent neural networks.

## Extent and Evaluation of Learning

Just like optimality property of plans is too hard to check and evaluate in terms of distance from being optimal (Long and Fox 2002), similarly domain model completeness and quality have no standard evaluation and analysis methods, and like the requirements specification, it cannot be objectively assessed, and proven correct. Learning systems and their output are typically evaluated empirically, based on their divergence from the reference model (which itself can be questionable from multiple perspectives by multiple experts). A step forward in defining the quality of DMs and to improve KE tools McCluskey et al (McCluskey, Vaquero, and Vallati 2017) uses the idea of DM as a formal specification of a domain, and considers what it means to measure the quality of such a specification. To build the notion of quality assessment, they used dynamic and static testing of the DM. (Vallati and McCluskey 2018) presents quality framework which aims at representing all the aspects that affect the quality of knowledge in DMs. The framework is based on the interaction between seven different sets that underpin

the domain quality.

In some cases, the extent of learning can be improved by accommodating or totally abolishing the inadequacies by the provision of richer observation samples of knowledge. On the other hand, for the learning of probabilistic domains, it is not easy to detect inadequacies, and on top, there are no current procedures to improve the stochastic actions when states that do not fit the model are observed.

The strength of the exploited learning approach or algorithm is the key factor to regulate the extent of learning by the system, as it may get stuck in local minima or not be able to capture patterns of the target knowledge within a reasonable time and memory requirements (Jiménez et al. 2012). For example, exploiting reinforcement learning method, to learn from reward-based approach can learn better in a stochastic environment as compared to the inductive learning (which is based on drawing from inference). Similarly, learning for conformant or contingent planning task, the suitable learning approach to adopt is by inference or by inductive generalization to find the best fit for the observed facts. The concept of model-lite planning (Kambhampati 2007) views a planning problem as an MPE (most plausible explanation) problem. These techniques search for solution plans that are most plausible according to the current DM, specifically for situations where the first bottleneck is getting the DM at any level of completeness.

## Learning Techniques and Algorithms

On the continuum of planning paradigms starting from the classical planning, learning/improving the domain theory is inconsistent with the classical planning assumption of a complete and correct domain theory (Zimmerman and Kambhampati 2003). Based on this fact, classical planning only supports offline learning of DM. Moving towards full scope planning, the real world problems present complex requirements like learning numeric variables, explicit time representation and derived predicates that enrich the descriptions of the world.

Among the majority of the DM learning techniques exploited for DM learning, much attention has been given to inductive (top-down approach) and analytical (bottom-up approach) learning by extrapolating from sample plans used as the evidence to make a probabilistic claim about all or most of the learned knowledge (Jilani et al. 2014). Analytical learning uses input additional knowledge such as initial/intermediate and goal states, fluents or other partial domain information as explanations of the collected example plans for deductive reasoning. Most of the knowledge extraction from the additional input knowledge used is based on direct liftings of the observed states in the input. Systems that learn very expressive DMs tend to demand most detailed input (Tate et al. 2012). Other inductive techniques such as decision tree learning, neural networks, and Markov Logic Network learning, have seen few learning applications. Numerous technique overviewed in this section is inclined towards full or at-least broader scope planning on the continuum. The overview is a non-exhaustive description of the relevant systems.

Many learning systems exploit the technique of Markov Logic Network (MLN) (Richardson and Domingos 2006) that applies the concept of a Markov network to first-order logic (FOL) and draw the inference from the evidence. The vertices of the Markov graph are taken as atomic FOL formulas, and the edges act as the logical connectives used to construct the formula. Several systems includeing (Zhuo et al. 2010) and (Zhuo and Kambhampati 2013) learn DM based on the idea of Markov network as a major driving approach. To deal with probability along with imperfect and contradictory knowledge, MLNs provide a dense language to determine very large Markov networks, and the ability to flexibly and modularly incorporate a wide range of domain knowledge into them (Richardson and Domingos 2006). LAMP system (Zhuo et al. 2010) uses the Markov Logic Network (MLN) technique to select the most likely subsets of candidate formulas from all the generated formulas which are later transformed into learned action models. It learns STRIPS action models (with quantifiers and logical implications) for classical planning from plan traces with partially observed states. TRAMP (Zhuo and Yang 2014) system conducts MLNs assisted transfer learning to learn DM.

NLOCM (Gregory and Lindsay 2016) presents a method of inducing cost (numeric) model for numerical planning. The approach operates from action traces, cost pairs and uses constraints programming based approach in order to identify the action parameter sets that influence action costs. It learns costs associated with important features of the object finite state machines, generated by the LOCM (Cresswell, McCluskey, and West 2013) family of algorithms. PlanMiner (Segura-Muros, Perez, and Fernandez-Olivares 2018) system requires in addition (potentially noisy) state information and presents a combined framework for learning the numeric and propositional components. (Lindsay et al. 2017) learns domain models from natural language action description.

Inducing DM and its features such as action duration for temporal domains has been well-studied using predictive modelling approaches of a relational decision and regression trees. Inducing regression trees is itself a well-known method to building models for numeric variables. Decision tree structures like a flow chart where learning takes place by labelling the variables at nodes and branches of the tree from the training data. Models in which state variables, represented in the form of trees can take discrete values are called classification trees while trees, where state variables can take continuous values, are called regression trees. Relational decision trees (Blockeel and De Raedt 1998) are the first-order logic upgrade of the classical decision trees (Jiménez, Fernández, and Borrajo 2013).

(Haigh and Veloso 1999) used regression tree learning (from execution traces of robot) to acquire rules that prioritize the activities of the robot ROGUE according to the values of its sensors. (Balac, Gaines, and Fisher 2000) used regression trees for DM learning by the robot through experience to make similar predictions under various environmental conditions in order to produce efficient plans. (Lanchas et al. 2007) automatically model the duration of ac-

tion execution as relational regression trees learned from observing plan executions. They use execution traces to infer important situational factors that lead to different execution times. Relational regression trees work with examples described in a relational language such as predicate logic. The system uses TILDE (Blockeel and De Raedt 1998) - a relational tree learning system that allows the construction of both relational decision and regression trees. (Jiménez, Fernández, and Borrajo 2013) presented *replanning when failure paradigm* to support planning in an uncertain and stochastic environment. The online learning component of the system allows it to learn probabilistic rules of the success of actions from the execution of plans and to automatically upgrade the planning model with these rules. Given the set of observations and real DM as input, the upgraded output DM is defined in PPDDL for probabilistic planning. To model and learn, non-deterministic action effects, (Pasula, Zettlemoyer, and Kaelbling 2007) developed a probabilistic, relational planning rule representation given a set of example action execution.

Due to the modelling-inherent complexity of hybrid domains, it is hard to obtain a model of the complex nonlinear dynamics that govern state evolution (Say et al. 2017). Unlike classical planning paradigm, research is still in its infancy about how the expressiveness in hybrid modelling may affect the hybrid reasoning in a dynamic world.

To effectively represent cascading of action effects in hybrid domain, Denenberg et al (Denenberg and Coles 2018a) designed three different methods for modelling sequences of continuous effects (on a single variable) that an action may cascade in a continuous domain. One of the methods is the PDDL 2.1 model using clips and durative actions and the remaining two are PDDL+ based. The authors have compared the performance of these models empirically using three different state-of-the-art PDDL+ planners and draw conclusions based on the performance of each of the model on two different learning problems. Alberto et al (Pozanco, Fernandez, and Borrajo 2018) present APTC, a urban traffic control system based on AP. The system assist DM design and adaptation according to continuous changes in the world by updating the model through monitoring and learning. The system learns the effects of the actions at a junction level and incorporates new actions in the DM.

In hybrid planning research, for past couple of years, researchers have been exploiting Deep Learning and other easily accessible tools such as TensorFlow to learn highly accurate nonlinear deep neural networks with least background knowledge of the model structure (Say et al. 2017). TensorFlow is an open source library for numerical computation and large-scale machine learning that in addition to other functions can simplify data acquisition process and models' training (Rampasek and Goldenberg 2016). In (Wu, Say, and Sanner 2017), authors exploited the use of TensorFlow in order to plan successfully in hybrid non-linear domains. Here the term "hybrid" is used in the context of MDPs (i.e. mixed domains of factored variables) as opposed to its meaning in PDDL+ planning community. The authors produced the results evidencing how Tensorflow can be highly scalable in terms of convergence and execution time.

It is worth noting that the problem of encoding DMs is being analysed not only from the point of view of generating models in a specific description language –such as PDDL– but also for generating different sorts of automatically exploitable models. Konidaris et al (2014) proposed a method for constructing symbolic representations for high-level planning by establishing a close relationship between an agent's actions and the symbols required to plan to use them.

## Considerable Issues

This section includes some considerable issues we have derived from our review.

The area of learning DM for planning has received proportionately less attention and narrow scope than the learning of search control knowledge for planning. One of the reasons for homogeneity and lack of diversification in DM learning methods and considered representation mechanism is the conflict between the configurations of learning approaches with planning approaches e.g. combining classical planning paradigm with increment DM learning.

Another related issue worth contemplating is about deciding which type of domain model or domain knowledge (over the continuum of planning) is hardest/easiest to learn and how hard/easy it is to get the corresponding training data. This might help researchers be more decisive about which part of the model is worth learning automatically with the available resources. Hybrid DM learning is a key to move on the spectrum of model abstraction from descriptive to more operational real-world DMs.

Another question worth pondering is the recommended sequence based on biases in which the considerable options for learning mechanism should be decided. Based on the complex interactions between the components of the learning system, finding the matching compatibility parameters among the components could be the right direction to answer the question. For example, to learn or refine the temporal DM for online planning system, what minimum input assistance would be required and which representation mechanism would best suit the purpose in order to balance the overhead cost incurred by the learning system so it doesn't overwhelm the gains in search efficiency.

Another abiding issue is the autonomous collection of rich observation samples of planning actions. There have been several plan recognition approaches (Ramírez and Geffner 2010) that can be exploited to output plan traces. Generally, there are three ways to collect example plans. The first is when plans are generated through goal-oriented solutions, the second through random walks and third through observation of the environment by a human or by an agent. Goal oriented plan solutions are generally expensive in that a tool or a planner is needed to generate a large enough number of correct plans to be used by the system. To do this one must also have a pre-existing domain model and problems with good coverage of world objects. Randomly generated plan traces usually under-sample the action and state spaces. Observation by an agent has a high chance that noise will be introduced in the plan collection; which can clearly affect the learning process. Currently, most working systems

assume the input knowledge to be correct and consequently not suitable for real-world applications. To increase potential utility, systems should be able to show equal robustness to noise.

There have been some recent work (discussed in Extent and Evaluation of Learning section) concerned with the creation of evaluation mechanism or matrix to judge the quality and robustness of DMs and declare it adequate for planners. As the sources of knowledge elicitation and model development are not mathematical procedures, therefore a DM cannot be measured on a correctness scale. To attract more research towards automatic DM acquisition, the 2009 edition of ICKEPS (Bartak, Fratini, and McCluskey 2010) was devoted to evaluating systems that involved the domain requirements to be captured in an application-oriented syntax for automatic or semi-automatic generation of a DM. However, the disconnect between research and application was highlighted also by the more recent 2016 ICKEPS competition, where the majority of competing teams used no informed method or supporting tools environment in the knowledge engineering process (Chrpa et al. 2017).

In order to gain researchers' interest in learning systems' development, current state-of-the-art learning systems should be more accessible and open to use by research students and the scientific community. Like planners, these systems should be available on-line and include an interface along with user guidance manual for ease of use by non-planning experts.

## Conclusion

At the state of the art, few automated techniques for learning DMs have been proposed; they differ in terms of considerable options discussed in this paper and so the accuracy and reliability of output DM. By this brief overview, we hope to draw research attention to the broader scope in learning efficient and expressive DMs for various planning paradigms by discussing common area to consider before the development of KE method for planning domains.

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
