# OpenReview forum: "Considerable Factors for Domain Model Learning in Planning"
_icaps-conference.org/ICAPS/2019/Workshop/KEPS — KEPS 2019_

### Official Review · AnonReviewer1 · 2019-05-08
**A survey of literature on learning methodologies for KE of planning domains**

**Rating:** 3
**Confidence:** 2

**Review:**

The paper presents a review of literature about learning methodologies as a support KE of planning domain models.

The paper is not easy to read, somehow verbose.

The topic in itself is adequate to the workshop, but I am not sure of the actual contribution provided here.

Excepted for the Section "Considerable Issues", the paper aims I think at organizing the existing literature, at presenting and commenting it and so on, to give some added value to the sum of surveyed works or to provide directions of future deployments.

But is this "added value" that it is difficult to grasp in this paper. I would suggest, at least in the presentation, to insist more on the "rational" behind the surveying activity, more than on the extensive description of surveyed works.

Some more specific comments:

- Abstract: please explain or remove 'AP' (defined only later in the Introduction)

- Introduction: "The correctness of the planner reasoning depends fundamentally on the quality of the domain knowledge..." This is too generic. I see what you mean, but I think correctness of planning relates to the capability of deriving correct plans wrt what is actually modeled in the domain, not correct plans wrt what the modeler wanted to model.

- p2, Temporal Planning: "Generally, the objective of modelling temporal constraints is a minimization of the plan makespan". Again too generic. Temporal Constraints are modeled for millions of reasons. To minimize the makespan can be an objective in some problems, but not the most important in general. In most of the cases, temporal constraints are modeled just because they exist for instance, and have to be satisfied. Or to express concurrency, resource allocation and so on...

- P3, Hybrid Planning. The planetary rover example is a bit "naive" like it is reported, even in such a generic context. The issue here comes mostly for the need of casting such a scenario into an action-base paradigma, not the most suitable for controlling such an asset. Check on the various literature around on the NASA's Mars2020 mission as an example, on the timeline-based planning approaches used in practice to control these rovers.

- p3,c1, r-7: expressivlely: maybe expressivity?

---

### Official Review · AnonReviewer2 · 2019-05-09
**Not a clear survey**

**Rating:** 1
**Confidence:** 3

**Review:**

The paper is about domain model learning in planning. It is not a research paper, there is no novel contribution. The paper can be best characterised as a survey, but honestly, I am not sure what is being surveyed. After reading the paper, I still cannot say what its purpose is. It is not a survey of domain model learning techniques, just a few of them are mentioned, no recent approaches such as  FAMA or LOUGA are covered, no critical comparison is presented. Actually, I have found many references quite old and only very few very recent works are referenced so there is a question of up-to-date content. The authors claim that the text is targeted to students, but I cannot recommend it to students as the text is very hard to follow. The structure is clutter, the used notions are not introduced, only experts can follow the text as the reader must be familiar with a lot notions used but never introduced in the paper. Just to give an example the text never says what domain model learning is.
I think the paper requires complete rewriting, the mission must be clear, the notions must be introduced to be accessible by non-experts (if targeted to students), the survey should be perhaps more focused (if length is restricted). What is added value? What does the reader take away after reading the paper?